# Cellulose Nanocrystals to Improve Stability and Functional Properties of Emulsified Film Based on Chitosan Nanoparticles and Beeswax

**DOI:** 10.3390/nano9121707

**Published:** 2019-11-28

**Authors:** Endarto Yudo Wardhono, Mekro Permana Pinem, Indar Kustiningsih, Sri Agustina, François Oudet, Caroline Lefebvre, Danièle Clausse, Khashayar Saleh, Erwann Guénin

**Affiliations:** 1Chemical Engineering Department, University of Sultan Ageng Tirtayasa, Cilegon 42435, Banten, Indonesia; mekro-permana.pinem@utc.fr (M.P.P.); indarkustiningsih@untirta.ac.id (I.K.); sriagustina@untirta.ac.id (S.A.); 2Integrated Transformations of Renewable Matter Laboratory (EA TIMR 4297 UTC-ESCOM), Sorbonne Universités, Université de Technologie de Compiègne, Rue du Dr Schweitzer, 60200 Compiègne, France; daniele.clausse@utc.fr (D.C.); khashayar.saleh@utc.fr (K.S.); 3Service d’Analyse Physico-Chimique (SAPC), Sorbonne Universités, Université de Technologie de Compiègne, Rue du Dr Schweitzer, 60200 Compiègne, France; francois.oudet@utc.fr (F.O.); caroline.lefebvre@utc.fr (C.L.)

**Keywords:** chitosan nanoparticles, Pickering emulsion, stability, water vapor resistant, mechanical strength

## Abstract

The framework of this work was to develop an emulsion-based edible film based on a chitosan nanoparticle matrix with cellulose nanocrystals (CNCs) as a stabilizer and reinforcement filler. The chitosan nanoparticles were synthesized based on ionic cross-linking with sodium tripolyphosphate and glycerol as a plasticizer. The emulsified film was prepared through a combination system of Pickering emulsification and water evaporation. The oil-in-water emulsion was prepared by dispersing beeswax into an aqueous colloidal suspension of chitosan nanoparticles using high-speed homogenizer at room temperature. Various properties were characterized, including surface morphology, stability, water vapor barrier, mechanical properties, compatibility, and thermal behaviour. Experimental results established that CNCs and glycerol improve the homogeneity and stability of the beeswax dispersed droplets in the emulsion system which promotes the water-resistant properties but deteriorates the film strength at the same time. When incorporating 2.5% *w*/*w* CNCs, the tensile strength of the composite film reached the maximum value, 74.9 MPa, which was 32.5% higher than that of the pure chitosan film, while the optimum one was at 62.5 MPa, and was obtained by the addition of 25% *w*/*w* beeswax. All film characterizations demonstrated that the interaction between CNCs and chitosan molecules improved their physical and thermal properties.

## 1. Introduction

Food packaging is an enclosing of food to protect it securely from outside influences, including chemical, biological, and physical effects. The packaging plays an important role in retarding food deterioration, retaining the beneficial effects of ingredients, keeping the quality and extending the shelf-life during distribution and storage [1]. Today, the flexible packaging material markets are dominated by plastic films, paper, and aluminium foil [2]. These materials have wide applications in the modern food packaging. There are several advantages of these materials compared to other materials. Aluminium foil for example, it has barrier functionality against the migration of moisture, oxygen, and other gases, and volatile aromas, and against the impact of light, it is generally better than any plastic laminate material [3]. The plastic films are mouldable, chemically resistant, lightweight, easy to process, and low cost. However, pure materials are rarely used by themselves [4]. To improve their performances, inorganic additives such as silica as a reinforcement, phthalate esters as a plasticizer, and bisphenol A (BPA) as a thermal stabilizer are added to these synthetic polymers [5]. Depending on the environmental conditions during their applications, some chemicals are potentially toxic when released from the plastic products and have adverse effects on the consumers [6]. Moreover, as non-biodegradable solid wastes, the discarded plastics contaminate natural, terrestrial water sources and marine habitats which are also threating to animals in the environment [4]. Numerous methodologies have been developed to solve the problems. Chitosan is a good alternative material to replace the use of conventional plastic as food packaging [1]. Chitosan is not only biodegradable but also edible which offers the possibility of obtaining thin films to cover fresh or processed foods to extend their shelf life. In addition, the thin films by themselves or acting as carriers of foods additives (i.e., antioxidants, antimicrobials), have been particularly considered in food preservation due to their ability to extend the shelf life [7,8,9].

Chitosan is a linear amino polysaccharide derived by deacetylation of chitin [10]. Chitin is the second most abundant natural polymer on the earth, a typical constituent of the cell walls of fungi; the exoskeletons of insects and crustaceans; and radula of mollusks [11,12,13]. Chitosan exhibits great properties, such as edibility, biodegradability, and biocompatibility [14]. However, this amino-polysaccharide shows low mechanical strength and poor barriers to moisture that limits its applications when compared to plastics [15]. To overcome the drawbacks, the addition of reinforcing materials is an effective technique in enhancing the mechanical properties, which bring to it new functional properties. Nano-sized particles could impart higher stiffness of the biopolymer. The smaller the filler particles, the better the percolation network within the matrix that will be obtained [16]. A different class of nanoparticles [17,18,19] has been used as reinforcement fillers. According to Dufresne, 2010 [20], cellulose nanocrystal, CNC, is a good material to get an appropriate percolation when incorporated in polymer matrices. Due to their large aspect ratio and ability to form interconnected network structures through hydrogen bonding, CNCs disperse within the matrix, and maximize the interfacial adhesion between the dispersed crystals and the polymer. CNCs are synthesized by removing the amorphous regions while keeping the crystalline regions through partial depolymerization of fibre sources [21,22,23]. To serve as a barrier against water vapor permeation, an extra hydrophobic component is added to the chitosan film [24]. A commercial beeswax (BW) is a favourable, eco-compatible material for the preparation of the film composites with a combination of chitosan. It has a high hydrophobicity characteristic and has been widely used in cosmetics, pharmaceutical, and food industries [25,26]. 

The major problem associated with compositing a hydrophilic polymer and hydrophobic materials is poor interfacial adhesion. Several strategies have been developed during the last few years, either bi-layer film lamination or emulsification [27]. In the bi-layer film, the hydrophobic layer is superimposed over the hydrophilic one; despite providing good barriers against gasses transfer, it tends to delaminate over time, develop pinholes, and exhibit cracks or even a non-uniform surface [28,29]. For these reasons, the bi-layer film is not popular in the food industry. Another approach is the emulsification, where the hydrophobic content is dispersed into the hydrophilic matrix. The homogeneity and the lipid droplets’ size is a determining factor the final property of the films [30]; however, the hydrophobic lipid naturally leads to a decrease in the mechanical strength of the films. During film formation or coating applications, solvent evaporation leads to a change in their microstructural properties, which induces the droplets coalescence [31], and then produces an apparent bilayer structure within the film [32,33]. The microstructural properties play a key role in the performances of the emulsion which depend on the preparation techniques, the types and quantities of constituents, and the compatibilities [30,34,35,36]. Manipulation of the interfacial property is a promising strategy to improve the microstructural stability of an emulsion. Pickering emulsion [37] is a kind of emulsion stabilized using solid particles. Compared to the one stabilized by surfactants, Pickering emulsion shows better performance because it requires less emulsifier and permits increased stability, and gives many other advantages [38,39]. A wide range of particles have been shown to be effective Pickering emulsifiers, such as silica [40,41], metals [42,43], cellulose [44], carbon nanotubes [45], clays [46,47], micro gels [48], and latex polymers [49,50].

The idea of this work was the development an edible composite film through the emulsification process with CNCs as the particle stabilizer and reinforcing filler simultaneously. The film was prepared through a combination system of Pickering emulsification and solvent evaporation. The oil-in-water emulsion was formulated by dispersing beeswax (BW) into colloidal chitosan nanoparticles containing CNCs with high-speed homogenizer. The chitosan nanoparticle (CNP) was synthesized based on ionic cross-linking with sodium tripolyphosphate (TPP) with glycerol as a plasticizer. The CNCs were added in order to facilitate the wettability of the particulate emulsifiers to interfaces but also improve the biocompatibility with CNP–TPP due to their similar chemical structures. The various properties were investigated, including surface morphology, stability, moisture barrier, mechanical strength, compatibility, and thermal behaviour. 

## 2. Materials and Methods 

### 2.1. Materials

Chitosan (low molecular weight grade, DD 75–85%) was purchased from Sigma–Aldrich, France; cellulose nanocrystals (CNC, in rod-like shapes with 90.94 ± 10.05 nm lengths and 12.58 ± 0.87 nm diameters) from CelluForce, Canada; sodium tripolyphosphate (TPP), Tween 80 and glycerol was obtained from Thermo Fischer Scientific, France; glacial acetic acid from Merck; commercial beeswax (BW) was acquired from PT Bronson and Jacobs, Indonesia. Demineralized water (conductivity of 0.06 mS cm^−1^) produced by a purification chain (Veolia, France) was used for all experiments. All the materials are used as a laboratory-grade without further purification.

### 2.2. Preparation of Hydrophilic Matrix CNP–TPP–CNC

Flow diagram of step by step film preparation and characterization is presented Figure 1:

#### Preparation of Hydrophilic Matrix CNP–TPP–CNC

The preparation of the hydrophilic matrix with a covalent linkage between CNP, TPP, and CNC consisted of three steps:
Chitosan solution, CS (1% *w*/*v*), was prepared by dispersing chitosan powder to 0.1 M of the acetic acid solution while continuously stirring at 300 rpm using magnetic stirrer at room temperature for 60 min. The resulting CS was then filtered using Whatman filter paper no. 1 to remove the impurities.Colloidal CNP–TPP was synthesized by adaptation ionic cross-linking method as described by Bao et al., 2008 [51,52]. The process was carried out under constant high-speed stirring at 10,000 rpm rate, 50 °C for 5 min using rotor-stator (POLYTRON PT-3100D-Kinematica, Swiss, Kloten, Switzerland). Glycerol, 10% *w*/*w* on a dry basis, was added previously to the CS as a plasticizer. The CNP was spontaneously obtained when TPP solution, 20% *w*/*w* on a dry basis of chitosan, was added drop-wise into the CS. Upon completion of the reaction, the cross-linked CNP–TPP was cooled at ambient temperature.Hydrophilic matrix CNP–T–CNC was carried out under ultrasonic irradiation using an ultrasonic processor (Vibra Cell, Type 72434, 100 Watts, horn diameter: 1.0 mm, Fisher Scientific, Illkirch-Graffenstaden, France). The CNC solution, 10% *w*/*w* on a dry basis of chitosan, was introduced into the colloidal CNP, and then kept at room temperature for 30 min under magnetic stirring (1000 rpm).

### 2.3. Preparation of Film Forming Emulsions

The hydrophilic matrix, CNP–TPP–CNC, was heated up at 75 ± 5 °C in a water bath to melt the BW lipid phase. The BW was weighed precisely (25% and 50% *w*/*w* on a dry basis of the film matter) and then introduced slowly in the matrix suspensions and homogenized at 14,000 rpm for 10 min using high-speed rotor-stator (POLYTRON PT-3100D-Kinematica, Swiss). The hot film-forming emulsion was quickly poured into hydrophobic Petri dishes to cast into films; then, cooled to room temperature in an ice bath. After that, the films were held in an air circulating chamber with constant relative humidity (50% RH) and temperature (20 °C) for 24 hours before peeling off the films. The transparent film samples were stored in desiccators at 60% RH for further characterizations. 

### 2.4. Nanoparticles and Film Characterizations

Morphology of the chitosan nanoparticle, CNP–TPP was observed using high-resolution transmission electron microscopy, TEM (JEOL-2100F, JEOL Ltd, Tokyo, Japan) and scanning electron microscopy, SEM (JEOL-2100F, JEOL Ltd, Japan). The sample was deposited on carbon-coated copper grids and the negative staining was achieved using uranyLess solution (Delta Microscopies, Mauressac, France). The size and diameter of the particles were measured by Image J (version 1.41 h) and origin pro-8 software.The stability of the oil-in-water emulsion was observed by bottle test for monitoring the extent of phase separation with time. The samples were introduced into glass tubes and then stored at elevated temperature (50 °C, 7 days) for the accelerated aging test. The volume percentage of the emulsion phase was reported as a function of time.Microstructural observations of the emulsions were visualized by an optical microscope (Leica DM2700, Leica Microsystems, Wetzlar, Germany), which was equipped with 3 magnification objectives 10 ×, 40 ×, and 100 ×. The sample was spread on a thin flat glass then observed under a high power objective of 100 × and then switched to the high power of 100×. The image captured was copied by the software Leica Application Suite, LAS.Wetting agent analysis on a solid surface was carried out by contact angle, CA, measurement using Drop Shape Analyzer (DSA 100; Kruss GmbH, Hamburg, Germany) with the static droplet method. One drop of deionized water was dropped on the surface of the film. The image was taken by a camera and analysed by software to obtain the CA.Mechanical testing was performed with ASTM D882-02. A tensile test machine (Strograph, Toyoseiki, Tokyo, Japan) was used to measure the tensile strength, TS, and elongation at the break, EB, of the film specimens (10 cm × 4 cm × 0.03 mm). The initial grip separation was set at 100 mm and cross-head speed at 50 mm/min. The TS was calculated by dividing the maximum load on the film before failure by the cross-sectional area of the initial specimen, while the EB was defined as the percentage change in the length of the specimen compared to the original length between the grips.The molecular structures of the film samples were studied by Fourier transform infrared spectroscopy, FTIR. The functional groups present in the nanoparticle CNP–TPP and its compatibility with CNC was determined using a Nicolet iS5 spectrometer (Thermo Scientific, Waltham, MA, USA). The measurements were taking 32 scans for each sample with a resolution of 4 cm^−1^, ranging from 400 cm^−1^ to 4000 cm^−1^ and a scanning speed of 20 mm/sat.The diffraction pattern of the films was investigated using X-Ray Diffraction, XRD, in a D8 Advance (Bruker, Billeric, MA, USA). Samples were examined with a scanning angle of 2θ from 10° to 40° at a rate of 1°/min with the CuKα filtered radiation.Thermal behaviour was characterized by differential scanning calorimetry, (DSC, Q100, TA Instruments, New Castle, DE, USA). In total, 30 mg of sample was placed in an aluminium crucible then inserted into the sample pan holder directly after weighing. The analysis was carried under constant nitrogen flow (50 mL/min), from −30 to 170 °C, at a heating rate of 5 °C/min.

## 3. Results and Discussion

### 3.1. Synthesis and Characteristization of COLLOIDAL CNP–TPP

In this work, ionic cross-linking with high-speed agitation (10,000 rpm) was performed to produce chitosan nanoparticles. The technique was modified from the original one [52]; our synthesis the temperature was set 50 °C rather than room temperature and the reaction time was 5 min rather than 3 min. This process offers a simple and mild preparation method for the synthesis of chitosan nanoparticles, which is based on electrostatic interaction between the amine groups of chitosan and negative charge of the TPP polyanion. The nano-scaled chitosan was obtained in an acidic medium, in which the optimum chitosan/TPP weight ratio was 4 compared to the original one, which was 5.

The morphological structure of the synthesized nanoparticles, CNP–TPP, was examined using TEM and SEM. The TEM image is presented in Figure 2a-1 and shows that the CNP–TPP was a sphere-shape even though its shape was not so regular. Most of the individual particles tended to form small agglomerates with average diameters of 13.4 ± 1 nm (Figure 2a-2). The nanoparticles look to be comprised of a dense structure constituted of interpenetrating polymer chains cross-linked to each other by TPP. Figure 2b displays the image of SEM that was captured at a high magnification. The structure of the CNP–TPP appears as spherical with a non-uniform particle size distribution and an agglomerated state. The modified reaction procedure tends to significantly reduce the size of chitosan nanoparticles from 60.6 to 13.4 ± 1 nm. According to Gokce, 2014 [53], the decrease in the particle size is maximum at the ratio of 4:1 where the binding of TPP with the ammonium group of chitosan is the most efficient.

### 3.2. Stability Study

To produce an emulsified-film composite, the emulsification of a hydrophobic material into a hydrophilic solution is necessary prior to film formation. The introduction of the emulsifier is intended to allow dispersion of the oil phase into the aqueous solution in order to improve stability and increase the oil distribution in the composite film.

#### 3.2.1. Forming Film Emulsion

In this work, the capability of emulsifier to stabilize forming film emulsions was studied by bottle test and optical microscopy. The oil-in-water (O/W) emulsions were prepared by dispersing various amounts of the BW phase (Ø = 25% and 50% *w*/*w* on a dry film matter) into the aqueous solution CNP–TPP with CNC as the emulsifier (1%; 2.5%; 5% *w*/*w* on a dry basis) and Tween 80 (2%; 5%; 10% *w*/*w* on a dry basis); then, they were stored at elevated temperature to carry out an accelerated aging test. The test was designed to increase the rates of potential chemical degradation or physical change for the emulsions and to get information on their shelf life in a relatively short time. The results are shown in Figure 3. 

Typical results of bottle test observation are given in Figure 3a. The histogram bar of the fresh O/W emulsion represents all sample emulsions after preparation (0 day, 20 °C); no phase separation was detected. After the aging test (7 days, 60 °C), an oil layer was visible at the surface of the sample emulsions that stabilized with CNC= 1% *w*/*w* which contained BW Ø = 25% and Ø = 50% *w*/*w* respectively. The emulsions that were stabilized with the higher one, namely, CNC = 2.5% and CNC = 5% *w*/*w*, produced stable emulsions for each BW content Ø = 25% and Ø = 50% *w*/*w*. No visual destabilization was observed. However, instability was pointed out after aging test for all samples that were prepared with Tween 80 (2, 5 and 10% *w*/*w*), whatever the BW percentage (Ø = 25% and Ø = 50% *w*/*w*). One must note that without CNC, the emulsion was not stable at all and quickly separated in two different phases.

The optical micrographs of the stable emulsions are represented in Figure 3b for samples of BW Ø = 25% containing CNC = 2.5% and Figure 3c for samples of BW Ø = 50% containing CNC = 5% *w*/*w*. At day 0, the BW droplets were spherical-shaped, homogeneously dispersed and in the range of 0.5–2 μm diameter-wise. After the test, the droplets remained unchanged but a small population of large droplets began to form in the sample of Ø = 50% (Figure 3c—7 days), and some droplet flocks also appeared in Ø = 25% (Figure 3b—7 days). However, both samples remained stable against coalescence and phase separation. These are results are in agreement with the results of bottle test, and therefore this constitution (25% and 50% *w*/*w* beeswax were dispersed in the hydrophilic matrix CNP–T–CNC containing 2.5% and 5% *w*/*w* of CNC, 10% *w*/*w* of glycerol) appears to lead to the best results for the emulsion formulation. Meanwhile, the emulsion which is prepared with Tween 80 (5% *w*/*w*) containing 50% *w*/*w* of BW shows a heterogeneous distribution of the BW droplets (Figure 3d—7 days). The number of small droplets decreased as coalescence occurred, which was followed by oil phase separation and sedimentation.

#### 3.2.2. Dried Emulsified-Film 

The stability within the dried emulsified-film was evaluated by the water contact angle (CA) measurement. The CA evaluation permitted us to study the stability of the film against moisture. The results of the measurement are presented in Figure 4. The films were formed at BW concentrations Ø = 25% and Ø = 50%, containing two types of stabilizer CNC (1%; 2.5%; 5% *w*/*w* on a dry basis) and Tween 80 (2%; 5%; 10% *w*/*w* on a dry basis). 

As presented in Figure 4a, for the lowest content of CNC (1% *w*/*w*), the CA of the dried film for sample Ø = 25% was 78.9° and Ø = 50% was 89.8°. The values increase with the increase of CNC at a constant BW concentration. The highest CA values were obtained for the highest CNC content (5% *w*/*w*) and reached 98.1° (Ø = 25%) and 106.3° (Ø = 50%), indicating an increased hydrophobicity and stability of the film structure against humidity. According to Perez-Gago and Krochta, 2001 [54], the smaller and the more homogeneous the oil droplets are in the emulsified film, the higher water contact angles are. In this case, the films display favourable surface properties and the Pickering particle of CNC are able to stabilize the emulsion during water solvent evaporation and film formation. The stability was promoted since the coalescence rate was decreased and the oil phase separation was inhibited when the water solvent evaporated [27]. The CA of the samples that was prepared with Tween 80 in the concentration range of 2%–10% *w*/*w* was smaller, under 70° (Figure 4b). It indicated that the films had a heterogeneous structure, which means that the films were composed of a hydrophilic continuous phase with inclusion of demulsified oil droplets due to coalescence and phase separation, resulting in the fact that the oil layer moved on the upper side.

In our case, the Pickering emulsion produced films that were more stable than the films prepared with an emulsion stabilized by Tween 80. The nano-sized CNC has a high surface area and high aspect ratio which probably allows for the formation of more stable emulsion, and therefore a more stable film than Tween 80 [55,56]. The solid nanoparticles are irreversibly adsorbed at the interfaces which produces a densely packed layer and forms an electro-steric protective shield around the droplets to prevent coalescence and phase separation [57,58]. There are many factors that influence the stability of Pickering emulsions, such as particle size, shape, and concentration, and surface wettability. Xiao 2018 [59] stated that the wettability of particles is the main parameter controlling the particle interfacial behaviours and emulsion stability. The diffusion of glycerol in the emulsion system can be used to optimize the wettability of the particles [60]. The presence of glycerol has a tendency to reduce the polydispersity of the emulsion [61]. The homogeneous dispersion of the oil droplets in the film structure indicates the effectiveness of CNC nanoparticles in adsorbing at the BW–glycerol interface which induces the stability of emulsion during the aging test and film formation which promote the improvement of water barrier property of the films [62]. However, the BW content also contributes reinforcement of the hydrophobic property of the dried film, as when its amount was increased from 25% to 50%, the CA was also improved which was strongly affecting water-resistance of the film.

### 3.3. Mechanical Properties of the Emulsified Films 

The effect of CNC loading on the mechanical properties of the emulsified film was evaluated. The mechanical properties are important to ensure the integrity of the film when referring to food packaging applications. Figure 5 displays the tensile strength, TS, and the ability to stretch (elongation at break, Eb) of the films which contained different concentrations of CNCs. 

Pure chitosan film exhibits features with low mechanical properties; namely, TS = 56.5 MPa and Eb = 7.4% (Figure 5-1); by reducing its size into nano-scale (CNP–TPP), we decreased the strength to 28.6 MPa, while the flexibility remained constant at 7.2% (Figure 5-2). However, introducing CNC into the CNP–TPP led to substantial improvement in the mechanical strength of the film composites, CNP-T-CNC. In this case, with the incorporation of 2.5% *w*/*w* CNCs, the TS reached a maximum of 74.9 MPa, whereas the flexibility increased to Eb = 20.0% (Figure 5-3). CNCs within the CNP–TPP film result in a great reinforcement effect. According to Mao, 2018 [63], the nanofiller of CNCs has a large aspect (length/diameter) ratio. The interaction between dispersed crystals cellulose and amino polysaccharide forms interconnected networks through hydrogen bonding which maximizes interfacial adhesion between the nanocrystals and the biopolymer molecules which improves the overall mechanical properties of matrix composite. The presence of glycerol in the matrix CNP–T–CNC weakens the inter or intramolecular force between the polymer chain, and thereby increases the elongation at break [64]. For the emulsified films, an increment of beeswax content negatively affects the film strength. By the same CNC loading of 2.5% *w*/*w*, the TS reduced to 62.5 MPa (Figure 5-5) as the BW was Ø= 25% *w*/*w*. The film became more brittle for the BW concentration higher than 25% *w*/*w*; a 50.9 MPa value was obtained at Ø = 50% *w*/*w* (Figure 5-8). Unfortunately, it was difficult to attain a higher mechanical strength by adding more than 2.5% *w*/*w* CNC. When 5% *w*/*w* CNC was incorporated, the TS was reduced to 49.4 MPa (Figure 5 and Figure 6) and to 39.5 MPa (Figure 5-9); thus, a compromise between the BW and the CNC content in the emulsified film was attained for BW Ø = 25% *w*/*w* and 2.5% *w*/*w* CNC. As it is expressed by TS measurements, introducing the hydrophobic phase lacks the structural integrity of hydrophilic biopolymer films. It caused a partial replacement of wax in the film matrix [65]. Concerning the film flexibility, the emulsification BW improved the Eb of the composite films; namely, to 26.6% for Ø = 25% *w*/*w* and to 27.8% for Ø = 50% *w*/*w*. It seems that the beeswax was able to form a continuous, cohesive matrix, resulting in increased elongation at break; even though the nanofiller CNC does not change elongation at break of the film, in this case, the value kept stable.

The parameters we tested suggest that the optimized film formulation may present better stability and mechanical properties. We made one containing: BW Ø = 25% *w*/*w* of dispersed phase and 2.5% *w*/*w* of CNC in the continuous phase.

### 3.4. Characterization of Emulsified Film 

After having prepared and evaluated emulsion properties, we produced films by slow evaporation of the water content at a controlled temperature and relative humidity. After 24 h the films were extracted and characterized using several analysis techniques: FT-IR, X-ray, and DSC.

#### 3.4.1. FT-IR Analysis

FT-IR spectroscopy was performed to detect the interaction between nanofiller CNC and chitosan nanoparticles in the composite films. The infrared bands were evaluated to determine the change in the chemical structure during the process; the results are presented in Figure 6 and the typical vibration bands are listed in Table 1. 

Figure 6a displays the vibration bands of amino polysaccharide of the chitosan which are characterized by the peaks at 960–1100 cm^−1^ which correspond to the C–O–C asymmetric stretch region; 1380 cm^−1^ was assigned to acetyl groups; 1658 cm^−1^ contributes to amide I groups; and 3342 cm^−1^ links to the stretching of an O–H bond. Figure 6b shows the CNP–TPP spectra: two peaks appeared at wave number 1640 cm^−1^ corresponding to amide I groups and at 1540 cm^−1^ corresponds to amide II groups. The vibration peaks in the range of 800–1300 cm^−1^ correspond to phosphate peaks of cross-linker TPP in the synthesized CNP–TPP [67]. Figure 6c shows the vibration bands of CNC. In general, the molecular skeletons of cellulose and amino polysaccharide of the chitosan are similar. The vibrations at 3340 cm^−1^, 2895 cm^−1^, 1430 cm^−1^, 1163 cm^−1^, and 895 cm^−1^ correspond to O–H bonding, symmetric C–H stretching, asymmetric C–H angular deformation, asymmetric C–O–C glycoside bonds, and asymmetric C–H angular deformation, respectively. The spectra of the composite film of CNP–T–CNC is presented in Figure 6d, in which the peak that was observed at 1258 cm^−1^ represents the asymmetric S=O vibration in CNC. It indicates that the crystalline cellulose had been successfully mixed to the cross-linked nanoparticles CNP–TPP. Some peaks were observed shifting to the lower wavenumbers, such as from 3342 to 3336 cm^−1^ corresponding to the O–H stretching band, and from 1658 cm^−1^ to 1650 cm^−1^ for the C=O stretching vibration. Meanwhile, the signal around 1590 cm^−1^ which was assigned to the bending of the N–H of amide II was broadened. A shift in wavenumber, intensity reduction, or broadening of the bands confirms the presence of non-covalent interactions in the composite films [68]. This implies that strong hydrogen bonds were formed between CNP and CNC molecules that, therefore, may improve the mechanical strength of the matrix composite CNP–T–CNC [69]. In Figure 6e no significant difference between the matrix CNP-T-CNC and emulsified film spectra could be noticed. The only changes that were detected when 25% *w*/*w* of beeswax was added to the matrix, were the more pronounced vibration bands at 2916 cm^−1^ and 2848 cm^−1^ that represent aliphatic –CH_2_ asymmetric and symmetric stretching vibrations [70]. These bands that correspond to the hydrophilic component are more intense compared to the CNP–T–CNC film.

#### 3.4.2. XRD Measurement

The presence of CNC and the nano chitosan was then confirmed with XRD measurements. Figure 7 shows the diffraction patterns of the pure chitosan, CNP–TPP, matrix CNP-T-CNC, and emulsified film composites in powder.

The X-ray diffraction pattern of the commercial shrimp shell chitosan is presented in Figure 7a. A broad diffraction peak found at 2θ = 20.6° corresponds to the amorphous state of the chitosan [71]. The characteristic diffraction peaks of the cross-linked nano chitosan CNP–TPP is displayed in Figure 7b. The pattern shows a broad diffraction peak centred around, 2θ = 19.8°, which is a typical fingerprint of the chitosan nanoparticles [72]. The lower intensity exhibited in the diffraction peaks was assigned to the amorphous state in nature. The spontaneous ionic cross-linking between the amino group of chitosan and polyanions of TPP led to the formation of the nanoparticles; this transformation increased the intensity of diffraction peaks of the CNP–TPP, while the peak slightly shifted from 20.6° to the lower one 19.8° and small peak appeared at θ = 21.6°. This can be attributed to the crystallized structure of CNP–TPP, for possible bonding interactions of the reaction of chitosan and TPP [73]. The XRD pattern of CNC (Figure 7c) was obviously detected at 2θ = 14.6°, 16.8°, 22.6°, and 34.1°; the pattern corresponds to the cellulose crystal I which consists of parallel molecular chains with a large number of hydroxyl groups enclosed in the crystal cells. They are closely connected with a great number of hydrogen bonds [74]. The characteristics of the XRD pattern from the matrix CNP-T-CNC are shown in Figure 7d. The new signals were observed at 2θ = 14.8°, 22.6°, and 34.1° that estimated as crystallite peaks of the CNC. It indicated that the inclusion of CNC into CNP–TPP was successful, which implies that the strong interactions between nano chitosan and crystal cellulose made the structure of the composite film more ordered which may have contributed to enhancing the mechanical properties of the films. The diffraction pattern of the emulsified beeswax film is shown in Figure 7e. The diffraction peak at 2θ = 23.9° for the emulsified film was found to be similar to the ones of beeswax at 2θ = 21.4° and 23° [75]. This pattern indicates that the composite film formed is a mixture of two materials rather than the formation of new material by chemical reaction [76].

#### 3.4.3. DSC Analysis

Thermal behaviour was carried out by DSC to interpret the physical characteristics of the composite films by measuring glass transition temperature T_g_ and crystalline melting point T_m_. DSC plot between heat flow and temperature for the samples is shown in Figure 8 and the summary is presented in Table 2. 

Figure 8a (blue line) and Figure 8b (red line) show the DSC thermogram of dried sample chitosan and CNP–TPP. The heat-flow of both samples displayed the same characteristics, including glass transition and crystalline melting temperatures. The signal of the chitosan revealed the glass temperature, T_g,_ to be 31.0 °C; the endothermic peak was distributed at 36.3–133.6 °C with the onset T_m_ = 81.4 °C. The crystalline melting temperature, T_m_ is also called dehydration temperature, which corresponds to the loss of the water content associated with the hydrophilic groups of chitosan [77]. At a higher temperature, it was observed that the heat flow was a monotonically increasing function of temperature, indicating that the sample was not completely dried with some bound water molecules, which were not removed. Compared to chitosan, the T_g_ of the sample CNP shifted slightly to the higher temperature to 36.1 °C, while the T_m_ detected moved to 1182 °C, which was then followed by the small inflection endothermic peak at 238 °C which that could be assigned to the crystallized structure during crosslinking with TPP. A shift in the T_g_ may be due to the interaction of chitosan molecules with TPP; since the shift was not too far, it can be assumed that the cross-linking did not affect the structure of the chitosan. The exothermic peak was assigned to monomer dehydration, glycoside bond cleavage, and decomposition of the acetyl and deacetylated unit [78]. Figure 8c (green line) presents the thermogram of CNCs. The signal showed a small inflection around 59.2 °C, which indicated the glass transition temperature, T_g_. The melting curve got to rise to a sharp peak around 260–290 °C with the onset at 288.6 °C. The effect of the 2.5% *w*/*w* of CNCs in the formulation, on the thermal properties of the matrix CNP–T–CNC is shown in Figure 8e (black line). The signal confirms that the same pattern to the one of the chitosan nanoparticles was observed. At low temperature, the T_g_ and the T_m_ were found to remain constant around 36.8 °C and 120.4 °C respectively. The addition of crystals of cellulose improved the heat resistance performance of the matrix which developed improved rigidity and crystallinity. The interaction among the components was stably bonded by electrostatic association and hydrogen bonds to rendered the breakage, dehydration, decarboxylation, and decarbonylation of glycosidic bonds, C–H bonds, C–O bonds, and C–C bonds in cellulose and chitosan molecules more difficult [63]. The DSC curve of emulsified film is shown in Figure 8e (brown line). It was observed that the peak obtained at 62.6 °C for beeswax was due to the solid-liquid transactions. The signal shows a little reduction in the melting point from 120.4 to 117.7 °C. 

## 4. Conclusions

This work described the fabrication of an emulsion-based edible film with chitosan nanoparticles as the polymeric matrix and CNCs as particle stabilizer and reinforcement material. The nanoparticles were prepared by a simple ionic cross-linking methodology under high-speed homogenization using TPP. The stable emulsion was obtained using beeswax rather than Tween 80. The CNC stabilizer lead to a percolating network and yielded a high stabilization at the oil–water interface occurring in Pickering emulsification. The FTIR and XRD comparison of the different constituents of the emulsion and of the films shows a proper integration of each component in the resulting film. The FTIR results provide supporting evidence that some specific non-covalent interactions between CNCs and nano-scaled chitosan are effective. Differential scanning calorimetry results reveal that the matrix had good thermal stability. The stable emulsion successfully allowed the production of films with interesting water resistance and mechanical properties. The addition of beeswax to the oil phase enhanced the water vapor barrier of the film. Mechanical properties of the emulsified film were strongly affected with tensile properties and elongation, which were deteriorated by the addition of beeswax. Therefore, these preliminary results show the potential applications of emulsified films, but the evaluation of other oil phases will need to be conducted.

## Figures and Tables

**Figure 1 nanomaterials-09-01707-f001:**
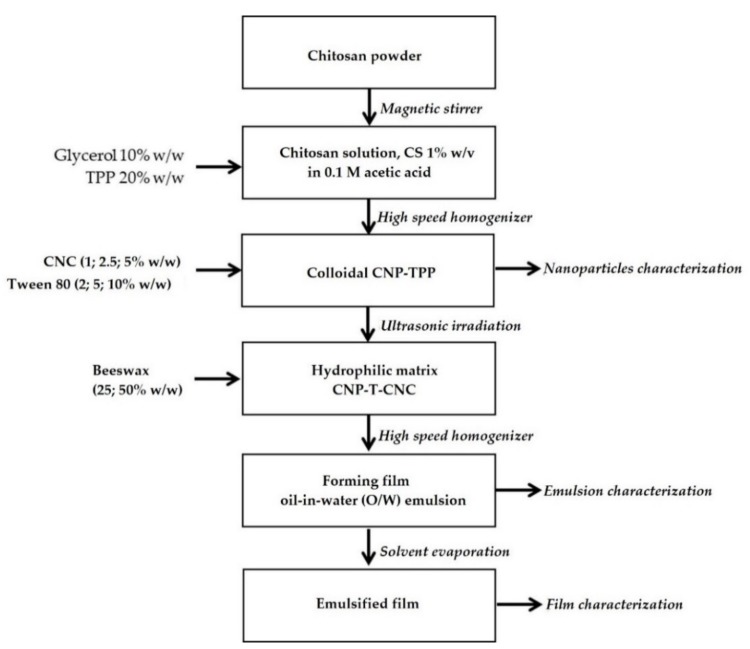
Flow diagram of the film preparation and characterization.

**Figure 2 nanomaterials-09-01707-f002:**
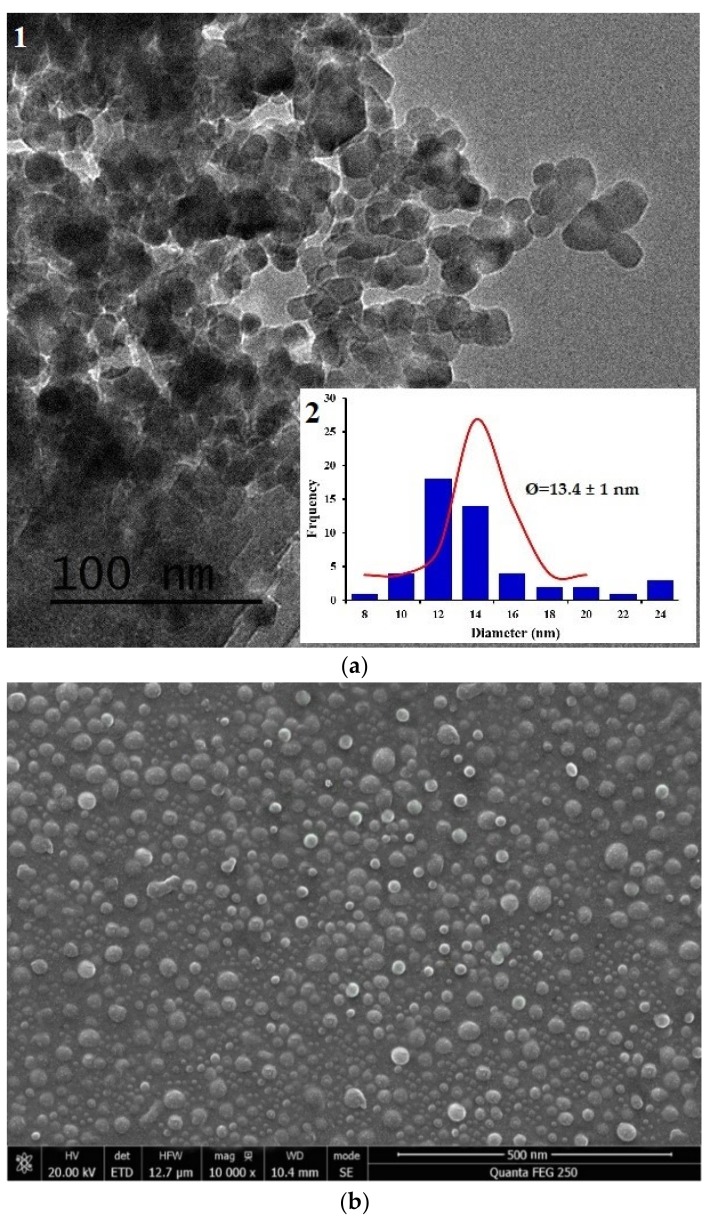
Morphological image of colloidal chitosan nanoparticle (CNP)–TPP was observed by: TEM (**a**-**1**); particle size distribution calculated by TEM (**a**-**2**); SEM (**b**).

**Figure 3 nanomaterials-09-01707-f003:**
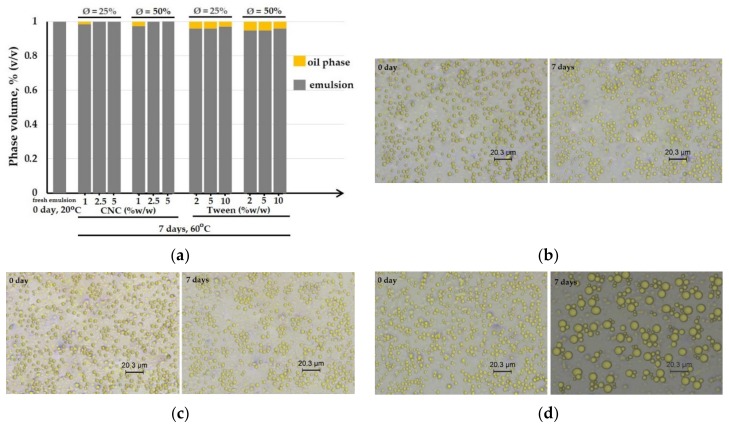
Bottle test observation of the emulsions (**a**). Optical micrographs of the emulsions stabilized with 2.5% *w*/*w* of cellulose nanocrystals (CNCs) before and after the aging test with: Ø = 25% (**b**); Ø = 50% (**c**). Optical micrograph of the emulsion stabilized with 5% *w*/*w* of Tween 80 before and after the aging test with Ø = 25% (**d**).

**Figure 4 nanomaterials-09-01707-f004:**
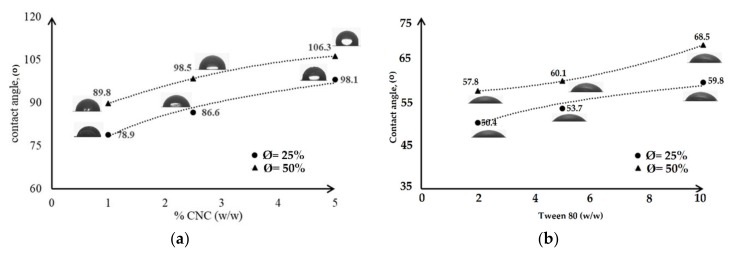
The effect of stabilizer content on the contact angle (CA) of the dried emulsified-films which stabilized: CNC (**a**); Tween 80 (**b**).

**Figure 5 nanomaterials-09-01707-f005:**
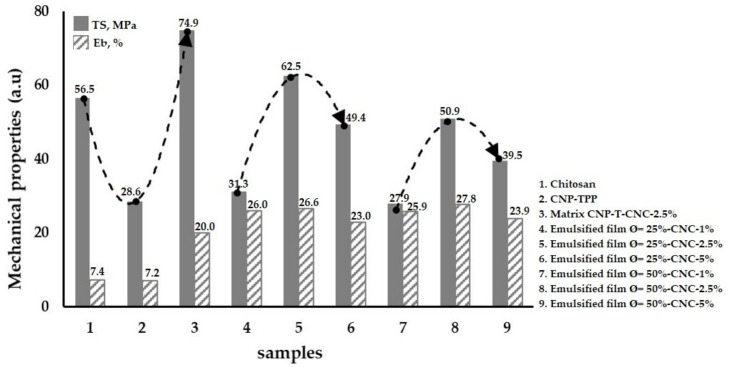
Mechanical properties of the chitosan-based film composites.

**Figure 6 nanomaterials-09-01707-f006:**
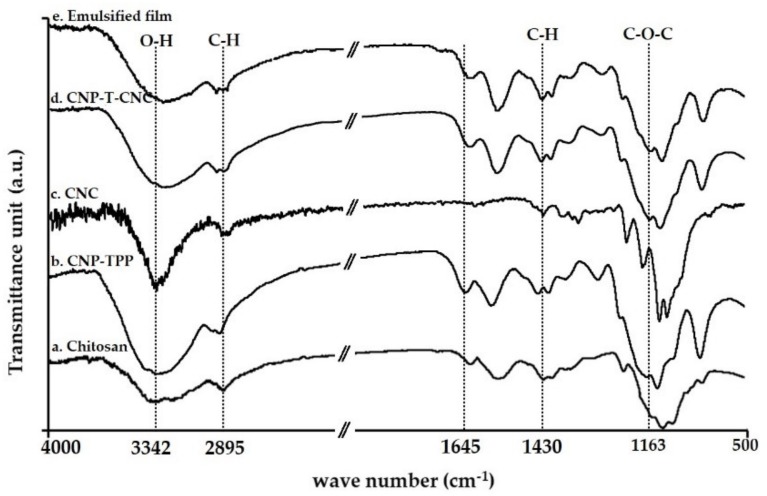
Fourier transform infrared (FT-IR) spectra of the chitosan-based film composites.

**Figure 7 nanomaterials-09-01707-f007:**
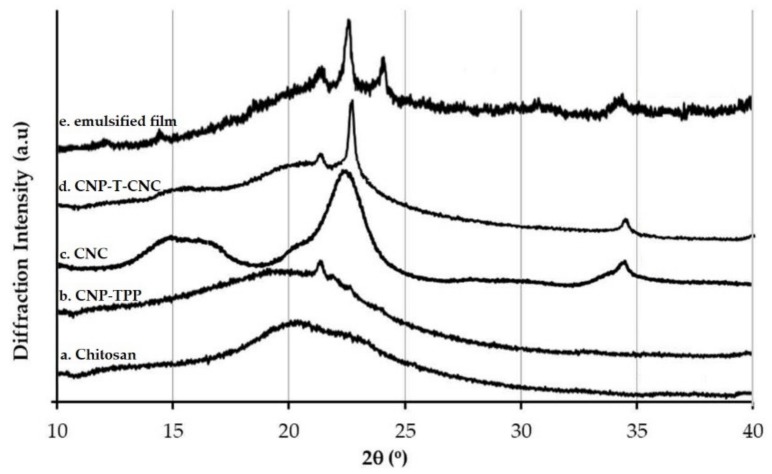
Characterization of chitosan-based materials by XRD.

**Figure 8 nanomaterials-09-01707-f008:**
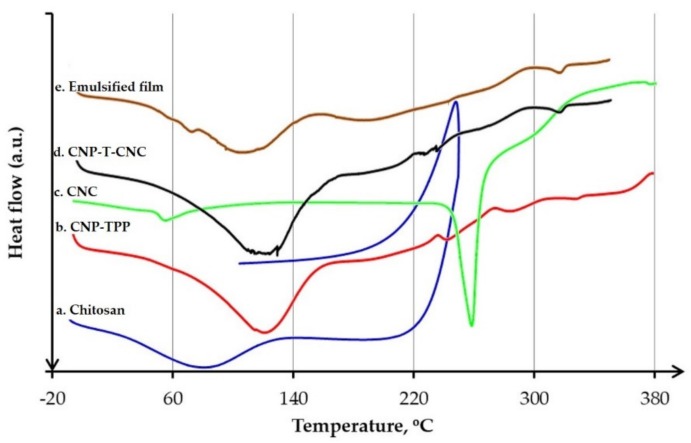
Characterization of chitosan-based materials by DSC.

**Table 1 nanomaterials-09-01707-t001:** Typical vibration bands for the Fourier transform infrared (FT-IR) spectra of the chitosan [66].

Wave Number, cm^−1^	Assignment
3342	Stretching of O–H bond
2895	Symmetric C–H stretching
1645	Stretching C=O amide I
1550	Bending N–H amide II
1383	Acetyl groups
960–1100	Asymmetric C–O–C stretching region

**Table 2 nanomaterials-09-01707-t002:** Characteristic thermal behaviour of chitosan-based materials

Temperature	Sample (°C)				
	Chitosan	CNP–TPP	CNC	CNP-T-CNC	Emulsified Film
T_g_	31.0	36.1	59.2	36.8	-
T_m_	81.4	118.2	288.6	120.4	117.7

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
