# Peer review of "Cellulose Nanocrystals to Improve Stability and Functional Properties of Emulsified Film Based on Chitosan Nanoparticles and Beeswax"

_nanomaterials, 2019, doi:10.3390/nano9121707_

Round 1
Reviewer 1 Report
It is well planned and methodologically correctly performed research. Authors choose proper characterization techniques that lead to constructive conclusions. The manuscript is well prepared from the editorial as well as the scientific point of view. Nevertheless, I found some points that need to be improved prior to publication, see below:
Please calculate the surface free energy on the basis of contact angle data. Authors wrote: Chitin is the second most abundant natural polymer on the earth, a typical constituent of the cell walls of fungi, the exoskeletons of insects, crustaceans, and radula of mollusks. – such statement needs to be confirmed by references, additionally, authors overlooked marine sponges and spider molts as a source of naturally prefabricated chitin. Please discuss and cite the following references:
10.1016/j.carbpol.2019.115301;
10.3390/ijms20205105;
10.1016/j.ijbiomac.2018.02.071
After improving of these points this manuscript will be suitable for Nanomaterials.
Author Response
Reply to reviewer #1 (file attached)

Reviewer 2 Report
I understand that the feature distinguishing the developed solution among other is that along with mechanical properties it is also edible. Introduction doesn`t give any benchmark on what Authors actually compete against. Are there any other edible foils/packaging that is on the market, is under development or are there any papers reporting such findings? Just very brief online search returned some interesting articles and patents. Some recent examples (there are papers on the filed from decades ago) are: https://doi.org/10.1080/02652039709374585, https://doi.org/10.1201/b19468. Some indeed are based on chitosan (https://doi.org/10.1111/jfpp.13090).
Section 3.1
Second paragraph in Section 3.1 (starting with “The morphological…”) seems like written by other person than Introduction. The language is worse and there are some grammar mistakes. Also:
500 nm is not magnification (p5l196). There is no information what was the size of CNP-TPP particles before modifying reaction condition, only claim that “The new reaction condition had given a significant effect in reducing the size of chitosan”. Inset (size distribution) in Fig 2 is too small and details are not visible.
Section 3.2
Why only 2.5% of CNC was tested? Was it somehow optimized? The title of the manuscript suggest focus on influence of CNC on stability of emulsions, but without more extended studies it is misleading. There are several ways to do this. For instance – take sample number 3 and vary amount of CNC from 0% (this is important control experiment that should be showed here) to some large value (at least 5%, maybe 10% (?)).
In p6l239 Authors say that the concentration of Tween80 was 2.5% w/w and refer to figure 3d. But caption to this figure says “5%w/w of Tween 80”. There are also other similar inconsistences throughout the manuscript. Please fix it.
Section 3.3.2
What does small peak at 21.6 degrees in CNP-TPP diffraction pattern correspond to? It is preserver in both CNP-T-CNC and emulsified films.
No control experiment for beeswax is shown, even despite Authors refer to it. Without it the claim in the last sentence of the section is not supported by data.
Section 3.3.4
Figure 6 – why Authors used a.u. for heat flow?
Hysteresis of blue line (chitosan) is not discussed.
Please indicate some arrows to the figure corresponding to claimed Td, Tm and Tg because it is very confusing in the current state. I can see why Authors claim change in Td upon addition of CNC to CNP-TPP from around 240 to 320 C, but I am not convinced that this is in fact a case. Are all the features above 150C reproducible between e.g. three separate experiments? From my experience these might be just fluctuations. It is risky to build theories on such observations. For instance, there is even some small peak at 220C in black curve, suggesting LOWERING and not INCREASE of Td.
Section 3.4.2
Why addition of CNC above 2.5% worsened mechanical properties of emulsified films? Was it the same for CNP-T-CNC matrix?
To sum up. I believe that the presented research is interesting but it is still in preliminary state (what Authors actually admit in Conclusion section). There are still some work to be done before publication.
Author Response
Reply to reviewer 2 (file attached)

Round 2
Reviewer 1 Report
The authors properly answered for my comments.
Author Response
Thank you for very much
Reviewer 2 Report
The manuscript is improved. I appreciate the explanation related to Figure 4.
For unknown reasons figure 2 was renamed to figure 1. There are two Figures 1 now.
There are still some minor editing errors, e.g. comma instead of dot separating decimal numbers
P1-305 "The nano-sized CNC has a high surface area and high aspect ratio" - I cannot see any evidence for high aspect ratio.
I am still not convinced by DSC resutls. Authors added Td, Tm and Tg marks but only in case of chitosan. I am not sure how they found Td in case of red, black and brown curves.
Author Response
The response is attached
